# Diagnosis of Epilepsy with Functional Connectivity in EEG after a Suspected First Seizure

**DOI:** 10.3390/bioengineering9110690

**Published:** 2022-11-14

**Authors:** João Matos, Guilherme Peralta, Jolan Heyse, Eric Menetre, Margitta Seeck, Pieter van Mierlo

**Affiliations:** 1Faculty of Engineering, University of Porto, 4200-465 Porto, Portugal; 2Department of Electronics and Information Systems, Ghent University, 9000 Ghent, Belgium; 3EEG and Epilepsy Unit, University Hospital of Geneva, 1205 Geneva, Switzerland

**Keywords:** focal lesional epilepsy, functional brain connectivity, machine learning

## Abstract

Epilepsy is regarded as a structural and functional network disorder, affecting around 50 million people worldwide. A correct disease diagnosis can lead to quicker medical action, preventing adverse effects. This paper reports the design of a classifier for epilepsy diagnosis in patients after a first ictal episode, using electroencephalogram (EEG) recordings. The dataset consists of resting-state EEG from 629 patients, of which 504 were retained for the study. The patient’s cohort exists out of 291 patients with epilepsy and 213 patients with other pathologies. The data were split into two sets: 80% training set and 20% test set. The extracted features from EEG included functional connectivity measures, graph measures, band powers and brain asymmetry ratios. Feature reduction was performed, and the models were trained using Machine Learning (ML) techniques. The models’ evaluation was performed with the area under the receiver operating characteristic curve (AUC). When focusing specifically on focal lesional epileptic patients, better results were obtained. This classification task was optimized using a 5-fold cross-validation, where SVM using PCA for feature reduction achieved an AUC of 0.730 ± 0.030. In the test set, the same model achieved 0.649 of AUC. The verified decrease is justified by the considerable diversity of pathologies in the cohort. An analysis of the selected features across tested models shows that functional connectivity and its graph measures have the most considerable predictive power, along with full-spectrum frequency-based features. To conclude, the proposed algorithms, with some refinement, can be of added value for doctors diagnosing epilepsy from EEG recordings after a suspected first seizure.

## 1. Introduction

Epilepsy is a neurological disorder characterized by recurrent seizures, which may be brief episodes that involve part or the entirety of the body, with interrupted loss or alteration of consciousness [1]. The World Health Organization (WHO) estimated in 2019 that more than 50 million people worldwide have epilepsy [2].

Epilepsy is usually diagnosed after a person has either experienced at least two unprovoked seizures (not caused by a known condition such as alcohol withdrawal or extremely low blood sugar) more than 24 h apart or when the patient experienced one unprovoked seizure with a high probability of recurrence [3]. Acute symptomatic seizures, vasovagal syncopes, patients with confusional states and other transient events are rather frequent in hospital emergency departments, amounting to large numbers of patients in whom epilepsy is suspected. It has been reported that epilepsy in adults has a misdiagnosis rate of 23% [4]. Errors committed in epilepsy diagnosis and seizure classification are known to lead to inappropriate decisions on the use or choice of anti-epileptic drugs and to other serious patient management errors [5,6].

More detailed classifications have been developed to describe the etiology and symptomatology of epileptic seizures, but in this study, we use the broader categories from [7,8,9,10]. In general, regarding its etiology, epilepsy can be divided into three groups:Idiopathic: genetically determined epilepsy, usually linked to a particular clinical characteristic;Lesional: usually related to a structural abnormality of the brain, an acquired condition;Non-lesional: no clear reason is present for the development of brain abnormalities.

Epileptic seizures can be further divided into two subtypes, considering the location of the brain where seizures start from:Generalized seizures: originated at some point in the brain, from where a widespread epileptic discharge rapidly involves the entire brain;Focal (partial) seizures: originated within networks that are limited to one cerebral hemisphere.

Appropriate diagnostic tools for epilepsy include electroencephalography (EEG), computerized tomography (CT), magnetic resonance imaging (MRI) and blood tests [5]. EEG and MRI are considered the key tools when it comes to diagnosis and treatment follow-up of epileptic patients [11].

EEG records the electrical field generated by neurons in the brain, with a frequency range of 0.5–100 Hz and a dynamic range of 2–100 μV [12], and is, therefore, able to expose physiological manifestations of abnormal cortical excitability that underlie epilepsy [13]. EEG is usually divided into five main frequency bands: delta (1–4 Hz), theta (4–8 Hz), alpha (8–13 Hz), beta (13–30 Hz) and gamma (30–100 Hz). Each one of them has distinct functional characteristics.

Since epilepsy is regarded as a structural and functional network disorder [14], functional brain connectivity may pose a role of relevance in the identification of patients with unprovoked seizures and a diagnosis of epilepsy. Functional brain architecture can be approached in two different ways [15]:Functional Segregation: identification of anatomical segregated and specialized cortical regions;Functional Integration: identification of interactions among functionally segregated brain regions, analyzing the communication of different brain regions.

Functional connectivity infers the relationships between the neuronal activity of several brain regions, investigating the statistical dependency between two or more time series [11].

Machine Learning (ML) is the field of Artificial Intelligence that automates analytical model building by creating systems that are able to identify patterns and make decisions from meaningful data [16]. ML algorithms have widespread applications in medicine and, in specific, in epilepsy, such as seizure prediction, detection and monitoring; epilepsy diagnosis and management; localization of the epileptogenic zone in focal epilepsy; surgical planning and prediction of surgical outcomes [16]. ML has been proven to be useful in several medical areas and shown to have a huge potential to support epileptologists’ decision-making. However, studies with larger and more diverse cohorts are still needed to bridge the gap to widespread acceptance in clinical practice [17].

Previous studies have found that functional brain connectivity computed from resting-state EEG without interictal epileptiform discharges can be altered in patients with focal epilepsy compared to healthy controls [18]. Further, global efficiency has been reported to be abnormal in resting state, and direct connectivity networks in focal epilepsy [19]. Furthermore, theta band functional connectivity has been found to be a sensitive predictor of epilepsy diagnosis (sensitivity 76% and specificity 62%) but on a relatively small dataset (*n* = 117) [20]. In another study, functional connectivity and network metrics were combined in a multivariate prediction model with high accuracy for children with partial epilepsy [21].

To the best of our knowledge, no other study has performed epilepsy diagnosis based on resting-state EEG from patients with a wide background of pathologies who arrived at a hospital emergency unit after a suspected first seizure. However, a six-center study has been carried out in the USA, Singapore and India, aiming to develop an automated epilepsy diagnostic tool from scalp EEG in a dataset of epileptic vs. healthy controls. The proposed system is based on the detection of interictal electrical discharges and a spectral feature-based classifier, with shown ability to aid clinicians in diagnosing epilepsy efficiently [22].

In this study, we hypothesize that functional connectivity derived from EEG may hold distinctive meaningful predictive power, which, allied to a well-designed ML classifier, can be of added value towards the diagnosis of epilepsy (versus other pathologies) in patients who had a suspected first seizure. With that goal in mind, we built a complete pipeline for epilepsy diagnosis, after a suspected first seizure, from resting-state EEG. The pipeline includes EEG preprocessing, epoch generation and cleaning, feature extraction, feature selection and the training of an ML classifier.

## 2. Methods

In this section, we will cover the steps and decisions taken in the pipeline designed for epilepsy diagnosis from resting-state EEG recorded in patients after a suspected first seizure.

### 2.1. Dataset Description

The dataset consists of routine, resting-state EEG from 629 patients who arrived at the epilepsy unit after a suspected first seizure, which provides a very interesting cohort of epilepsy patients vs. other pathologies with similar transient events. A first seizure event—which could be epileptic or not—is characterized by a paroxysmal transient neurological deficit. The patients included in this study arrived consecutively in the epilepsy unit from 2010 to 2017. We excluded all patients under 16 years old, all patients with evident acute symptomatic seizures and all cases immediately identified as non-epileptic.

The EEG signals were recorded at a 256 Hz sampling frequency, have an average length of 17 min ± 6 min (EEG lengths range between 10 min and 1 h, with a median EEG length of 16 min) and were recorded with standard 10–20 montage, with 19 channels. Hyperventilation and photic stimulation were present in the dataset; since this is only during a short fraction of the recording and equally present in all recordings, we assume that the effect on our analysis is negligible.

The ground truth for the diagnosis of patients was determined after a medical follow-up of 2 years, based on further exams and antecedent knowledge besides EEG, even though the differential diagnosis of epilepsy is prone to errors. For that, the International League Against Epilepsy (ILAE) criteria (2017) were applied [23], suggesting that epilepsy (disease and not epileptic seizure) can be diagnosed either if a single seizure occurred with a risk of relapse over 60% without treatment (mainly when an epileptogenic brain lesion was present, or interictal epileptic discharges at the EEG were identified) or if the patient experienced a second event.

Therefore, the proposed classifier may be able to have more correct diagnoses after the first examination, thus eliminating long periods of uncertainty and wrong drug intakes for the patient. In addition, the metadata also includes the “EEG result”, which is the clinical evaluation of the first EEG, which can be “normal” or “abnormal”, the latter being related to some pathology in the EEG. It should, however, be noted that this label of EEG abnormality is not specific to epilepsy but rather indicates the presence of any disease-related abnormality in the EEG signal. This label will be used as a rough benchmark for our classifier performance.

The patients who were sleeping during EEG recording (57 subjects) were excluded based on visual interpretation by the electrophysiologist since sleep EEG is known to present changes in brainwave patterns [24]. Uncertain diagnosis labels were also discarded (68 subjects). Regarding the remaining 504 patients’ metadata distribution, which is presented in Figure 1, the dataset is quite balanced with regard to gender, age, and diagnosis. A total of 56.6% of the patients are males, 43.4% females; 57.7% were diagnosed with epilepsy, 42.3% are non-epileptic. Of all epilepsy patients, only a small portion have generalized idiopathic (24) or focal non-lesional (48) epilepsy, while the largest group of epilepsy patients have focal lesional epilepsy (219). Because of this imbalance, we will, later in this study, also train a classifier aimed at distinguishing patients with focal lesional epilepsy from non-epileptic patients.

This study was reviewed by a certified epileptologist for clinical aspects (MS). The procedure of the study was approved by the local ethical committee of Geneva.

### 2.2. EEG Preprocessing

EEG preprocessing was implemented with MNE-Python, an open-source Python package for the analysis of human neurophysiological data [25].

A symmetric linear-phase finite impulse response (FIR) pass-band filter, with a Hamming window of 6.6 s, was applied to the entirety of the raw signal. To filter the EEG signal to its frequency range of interest, the lower pass-band edge was set to 1 Hz and the upper pass-band edge was set to 40 Hz.

To automatically remove electroencephalogram (EOG) artifacts from the EEG signal, such as the ones caused by eye movement or blinking, independent component analysis (ICA) was implemented. ICA is widely used for blind source separation in EEG signal processing since it is able to isolate signals from different sources [26]. A total of 15 components were computed for each subject’s EEG. From a subject with clear and clean EOG components, two independent components were obtained using ICA and manually selected as a template for EOG artifacts (Figure A1 of Appendix A). Only independent components that were clearly and solely related to EOG artifacts were selected. Since artifact patterns are similar enough across subjects, in the remaining dataset’s subjects, highly correlated independent components (above a threshold of 0.8) were dropped, thus removing the majority of EOG artifacts in all subjects.

Finally, non-overlapping epochs were generated. Previous studies show that longer epochs may fail to capture functional brain connectivity due to EEG’s non-stationary nature [27]. To combine both the necessary low-frequency resolution for band power features and the effect of longer epochs [28], two different sets of epochs were created: for band power features, epochs with 256 Hz sampling frequency and 5 s length; for functional connectivity-related features, epochs were 2.5 s long and the EEG signal was downsampled to 128 Hz to minimize the computational cost of further processing. To remove zero-signal segments, epochs with less than 1 μV peak-to-peak voltage were excluded based on EEG visual inspection.

### 2.3. Epoch Cleaning

Bad epochs and sensors were automatically removed using the AutoReject library [29]. By setting sensor-specific thresholds through 10-fold cross-validation, the rejection of noise outliers, correction of bad channels with interpolation and removal of epochs with many corrupted channels was performed, as one can observe in Figure A2 of Appendix A.

### 2.4. EEG Bipolar Referencing and Scalp Groupings

Bipolar montages have been shown to avoid redundant information amongst channels since only potential differences between adjacent electrodes are recorded [11]. To reduce the effects of volume conduction, which can play a large role in sensor space, a longitudinal bipolar referencing system was set.

Aiming to find significant differences among different brain regions, four scalp groups were defined: anterior-left (AL), anterior-right (AR), posterior-left (PL) and posterior-right (PR), as used in previous studies [30]. The anterior groups include the frontal and temporal lobes, while the posterior groups include temporal, parietal and occipital lobes. Though there is some channel overlap, with electrodes Fz-Cz and Cz-Pz simultaneously belonging to left and right groups, it was considered beneficial to keep them since they might hold important brain information. The longitudinal bipolar montage and the four proposed scalp groups may be found in Figure 2.

### 2.5. Epoch Selection

To reduce the computation cost of dealing with all subjects’ epochs, in the feature extraction step, only the 50 best epochs were selected, depending on the information to be extracted from them. The number 50 was chosen in a trade-off between the available computational power while still using enough epochs to properly describe the patient’s data. The selection process takes into consideration the band powers and maximization of frequency content, given the frequency band of interest. For instance, for any alpha band-related feature, the selected epochs were the top 50 in terms of relative alpha band power. This way, we obtain the epochs that best represent the alpha activity.

Additionally, whenever the feature to be extracted concerned all frequencies (and bands), 1–30 Hz, and not only a specific frequency band, the assessment was based on a global band power (*GBP*) measure. This measure consists of the total absolute power of all bands over the variance between bands (Equation (Equation 1)). The selected epochs will have the frequency power maximized, and the variation across relative band powers minimized, representing all bands evenly.
(1)$$GBP=\frac{TotalAbsolutePower}{BandPowerVariance}$$

The computation of band powers was performed with YASA Python toolbox [31], which implements the Welch method [32]. To tackle the 1/f decay, the following modification was implemented: each frequency power was multiplied by its respective frequency, thus normalizing the frequency powers across bands. The final measure was the relative band power, which expresses the power in a frequency band as a percentage of the total power of the signal. Each epoch is assessed based on an average through all its channels’ band powers.

### 2.6. Feature Extraction

#### 2.6.1. Band Power Features

Since there has been a description of alterations in EEG’s band powers during different epileptic states [33,34], delta, theta, alpha and beta band powers were considered relevant features to include in the pipeline. Other studies report activity occurring during interictal periods in different frequency bands (up to 250 Hz) [35], which were not considered in this study.

Only the 50 selected epochs of each band were considered for the computation of the respective relative band power. In each epoch, the mean relative band power across all electrode pairs was taken. In each subject, all epochs were combined, per band, with the median and standard deviation. With the set of epochs selected for the global band, the mean total absolute power was also computed and included as a feature.

#### 2.6.2. Functional Connectivity

Functional connectivity studies temporal correlations amongst spatially distinct neurophysiological events [11]. Using datasets from different techniques, such as EEG or functional magnetic resonance imaging (fMRI), functional connectivity examines the regional interactions in the brain on a macroscopic level [36].

Functional connectivity measures can be either directed or undirected, linear or nonlinear connections in the time or frequency domain. The measures are based on the amplitude or the phase and are either bivariate or multivariate [11].

From several existing measures of functional connectivity, the selected ones in this study are from four different categories, thus maximizing the information to be extracted from EEG. From these four categories of connectivity measures, we selected one measure for each: from Correlation and Coherence-related measures, the Imaginary Part of the Coherency (IMCOH); from Phase Synchronization measures, phase-locking value (PLV); from Information-based measures, Mutual Information (MI); and from Granger Causality measures, the Partial Directed Coherence (PDC).

*IMCOH*: undirected and bivariate measure that is based on the Cartesian representation of the coherency. It has been reported to be quite insensitive to volume conduction artifacts, as well as robust for the identification of te interaction between brain regions, therefore, being a good way of representing brain interactions [37]. *IMCOH* is defined in Equation (Equation 2),
(2)$$IMCOH=\frac{Im(E[Sxy])}{\sqrt{E[Sxx]*E[Syy]}}$$
where *E*[] denotes average over epochs, *Sxy* the cross-spectral density between both EEG signals, *Sxx* and *Syy* the power spectral density of the first and second signals, respectively. The latter was computed with the multitaper method [38].*PLV*: measures the variability of the phase between two signals, being amplitude insensitive and depending only on the phase relation between both signals. It is a directed, bivariate, nonlinear measure based on a signal’s phase and is time-dependent [39]. It is calculated with Equation (Equation 3).
(3)$$PLV=|E[\frac{Sxy}{\left|Sxy\right|}]|$$*MI*: an information-based measure that evaluates the undirected nonlinear dependence between two signals. *MI* is bivariate, as well as amplitude-based and time-dependent [11]. It can be computed as Equation (Equation 4) suggests,
(4)$$MI_{XY}\;=\;\sum_{x,y}^{}\left(p_{XY}\left(x,y\right)\;log\left(\frac{p_{XY}\left(x,y\right)}{p_{X}\left(x\right)p_{Y}\left(y\right)}\right)\right)$$
with $p_{X}\left(x\right)$ and $p_{Y}\left(y\right)$ representing the probabilities corresponding to signals *X* and *Y*, respectively, and $p_{XY}\left(x,y\right)$ the joint probabilities of *X* and *Y*.*PDC*: based on autoregressive models (Granger Causality measures). A multivariate autoregressive model (MVAR) models K signals as a linear combination of their own past, plus additional white noise, as Equation (Equation 5) shows.
(5)$$X\left(n\right)\;=\;\sum_{m=1}^{p}A\left(m\right)X\left(n-m\right)+E\left(n\right)\;$$*X(n)* is the signal matrix at time *n*, *E(n)* is the matrix containing the uncorrelated white noise at time *n*, *p* is the model order and *A(m)* is the K × K coefficient matrix for delay m. *A(m)* estimates the influence of sample *x*(*n* − *m*) on the current sample *x(n)*, to which a Fourier transform is applied. PDC is computed with Equation (Equation 6).
(6)$$PDC_{ij}\left(f\right)\;=\;\frac{A_{ij}\left(f\right)}{\sqrt{\sum_{k=1}^{K}{\left|A_{kj}\left(f\right)\right|}^{2}\;}}$$*PDC* is a measure that is directed, multivariate, linear, amplitude-based and computed in the frequency domain. It is normalized in respect to the outgoing flow, which equals one at each frequency, and it exclusively shows the direct interrelations between the signals [11].

While IMCOH conveys the Cartesian representation of the coherency, which is regarded to be more representative than the magnitude and phase of brain interactions [40], the PLV is insensitive to the magnitudes of the signals and only depends upon the phase relationship between two signals [11]. On the other hand, MI can describe nonlinear patterns, providing complementary information. Many crucial neural processes are known to have nonlinear patterns [41]. Finally, PDC takes into consideration directed statistical dependence among multiple signals, taking a look at past samples and causality [42].

Three different toolboxes were used to compute the previously mentioned functional connectivity measures. IMCOH and PLV were computed using the MNE-Python toolbox [25], MI was computed using Scikit-Learn toolbox [43] and PDC was computed using SCoT toolbox [44].

IMCOH, PLV and PDC are spectral features, and, therefore, their values are frequency-dependent. For each of the five defined bands (delta, theta, alpha, beta and global), a connectivity matrix is computed using the 50 selected best epochs for the band of interest. An important note for this section is that the delta and global frequency bands suffered a slight change, regarding their range, which became 2–4 Hz and 2–30 Hz, respectively, due to the necessary minimum of five cycles in lower frequencies, given the epoch length of 2.5 s, for the proper computation of spectral connectivity. As for MI, which is computed in the time domain, the used set of selected epochs is the one based on the broadest frequency spectrum, the global band.

Regarding IMCOH and PLV, frequency bins of 0.4 Hz are used on each epoch within each band’s frequency range. The values of all the frequency bins are combined by computing the mean value within a given band. To get a single value per band and measure, for each of the source’s pairs, epochs are combined originally by the MNE toolbox with the mean. In the case of IMCOH, absolute values are taken into consideration.

In the case of MI, since the combination of epochs from the same subject was not originally implemented by the Scikit-Learn toolbox, the median was preferred over the mean. The median was considered to be less sensitive to outliers and, therefore, a possibly better descriptor of central tendency.

When computing the PDC, several aspects are to be taken into consideration. Even though the SCoT toolbox proposes the use of PCA and ICA to mix cortical sources [44], this step is bypassed by setting mixing and unmixing matrices equal to the identity matrix, and each EEG sensor is taken as a unique source. The MVAR model has no Ridge penalty, and its order is set to 8, optimized by cross-validating the mean-squared generalization error. For each subject, all epochs and channels are combined in a single MVAR model (Equation (Equation 5)), from which the PDC is computed afterward using Equation (Equation 6). Frequency bins are in the 0.25 Hz and range from 0.25 to 30 Hz. For each frequency band, the PDC values on that particular set of frequency bins are combined using the mean.

Finally, connectivity matrices were summarized per brain region, according to the already defined scalp groupings: AR, AL, PR and PL, in Figure 2. For each group, from the channels of interest, the mean value and its standard deviation were computed, thus achieving a single mean and std per scalp group rather than a matrix with all connections.

#### 2.6.3. Graph Measures

To summarize functional connectivity measures and provide further predictive power, graph measures were computed from connectivity matrices.

The degree is a measure that illustrates how many links are connected to a given node. Individually, this measure indicates how important a given node is in the network, but the degrees of all the nodes in a network reflect how developed and resilient a network is. The chosen metrics to measure the degree were the *in* and *out degrees*—directed measures that reflect the number of inward and outward links on a node, respectively—and the *node strength*, defined as the sum of the weights from all adjacent links [45].

To evaluate functional segregation, which analyzes the potential of occurrence of specialized processing in deeply interconnected brain region groups (clusters), the chosen metric was the *clustering coefficient*, which reflects the existence of clusters of connectivity around individual nodes [45].

Another group of graph measures that were considered was functional integration. This measure allows a quick combination of specialized information from distinct brain regions, evaluating how easily brain regions communicate and the information routes between brain regions, known as paths. From this group of measures, the *efficiency*, which represents the average inverse shortest path length among the network’s pairs of nodes, has been characterized by some authors as a superior measure of integration [46] and has, as a consequence, been the chosen functional integration metric.

Finally, centrality measures were also included, which assess individual nodes’ importance in the most important brain region’s interactions, facilitation of functional integration and resilience to insult. The fraction of all the shortest paths that pass a network’s given node is known as *betweenness centrality*, which was the selected centrality measure [45].

For each connectivity matrix, frequency band, scalp group and functional connectivity measure, the following graph measures were computed using the brainconn toolbox [47]:In and Out Degrees: these measures are computed only for the PDC, as it is the only directed functional connectivity feature being used. From each scalp group’s matrix, only the maximum values are considered.Node strength: after the computation of each channel’s node strength in IMCOH, PLV and MI matrices, the mean and standard deviation values are computed for each scalp group. However, for PDC, only the mean is computed here, as the standard deviation would be strongly correlated with the maximum value of in and out degrees.Clustering Coefficient and Betweenness Centrality: just like the node strengths, the mean and standard deviation are calculated from each connectivity matrix for all functional connectivity features.Efficiency: the computation of this measure results in one single value for each matrix regarding one subgroup, band and functional connectivity.

#### 2.6.4. Asymmetry Ratios

Furthermore, a studied hypothesis was the existence of distinctive asymmetries between the left and right hemispheres, which was performed by comparing anterior-left vs. anterior-right and posterior-left vs. posterior-right features. These asymmetries were computed as a ratio between the described groups of each of the different graph measures, using the mean or maximum value of the graph measure’s matrix, depending on the graph measure, as described in the previous section. All the asymmetry ratios are non-sensitive to lateralization, such that the maximum value between the two groups being compared was always the numerator, guaranteeing a value ≥1, as Equation (Equation 7) suggests.
(7)$$AsymmetryRatio=\frac{max(left,right)}{min(left,right)}$$

To synthesize the feature extraction, one can find all features that were input to the feature selection step in Table 1.

### 2.7. Data Split

Data (features and labels) was split into training and test sets in an 80/20 ratio. Models were trained with 5-fold stratified cross-validation, preserving data distribution in every split.

### 2.8. Dimensionality Reduction of Feature Space

Different methods for feature selection or dimensionality reduction of the features’ space were trained together with the ML models. The trained methods included: univariate feature selection of K best features, with ANOVA F-value for feature scoring [48]; Principal Component Analysis (PCA) [49]; Elimination of highly correlated features. In the latter method, a correlation matrix can be computed from all training features and for each pair of features with a correlation above a determined threshold, one of the highly correlated features is removed. The threshold is to be optimized in the training process, and, in this study, it was set to 0.92.

### 2.9. Machine Learning Classifier

Several models were trained and assessed in validation sets toward the design of a final classifier. The considered ones were Support Vector Machine (SVM) [50], Multi-layer Perceptron (MLP) [51,52], Random Forest Classifier (RFC) [53,54] and Logistic Regression (LogReg) [55]. All ML models were implemented using Scikit-Learn’s toolbox [43]

The features were normalized using Equation (Equation 8),
(8)$$NormalizedFeatureValue=\frac{x-\mu }{\sigma }$$
where *x* stands for the feature value, μ for the mean feature value and σ for the features values’ standard deviation. In the case of MLP, the features were also set between 0 and 1.

For result assessment, the following metrics were used:*Accuracy*: measures how often the algorithm classifies a data point correctly, defined in Equation (Equation 9),
(9)$$Accuracy=\frac{TP+TN}{TP+TN+FP+FN}$$
where *TP* is a true positive, *TN* a true negative, *FP* a false positive and *FN* a false negative;*Sensitivity*: measures the proportion of positives that are correctly identified, also known as true-positive rate (TPR), defined in Equation (Equation 10);
(10)$$Sensitivity=\frac{TP}{TP+FN}$$*Specificity*: measures the proportion of negatives that are correctly identified, also known as false positive rate (FPR), defined in Equation (Equation 11);
(11)$$Specificity=\frac{TN}{TN+FP}$$Confusion Matrix: a table in which each row represents the instances in an actual class, while each column represents the instances in a predicted class [56];Receiver Operating Characteristic (ROC) Curve: plots the TPR against the FPR at various threshold settings at which the decision is taken. The optimal cut-off point for a model will be the one that maximizes TPR and minimizes FPR, being used for the computation of the optimal confusion matrix. Surrogate predictions were generated by taking the 95th percentile (for statistical significance) from 100 permuted sets of predictions [57];Area under the ROC curve (AUC): it measures the area underneath the ROC curve, ranging from 0 to 1. AUC provides an aggregate measure of performance across all possible classification thresholds [58], being the preferred ML evaluation criterion in this work.

Different classification tasks were then performed with the extracted features toward an accurate binary distinction between epileptic and non-epileptic patients.

In a first **generalist approach**, the dataset was simply split according to diagnosis labels from medical annotations: patients diagnosed with epileptic seizures were considered epileptic; patients diagnosed with any other pathology after the suspected first seizure were regarded as non-epileptic.

The classification task was then focused on a more specific group, with **focal lesional epilepsy vs. non-epileptic classification**, i.e., only one type of epilepsy was considered. In a more specialized approach, the patients were segregated by their gender and age, which generated four subsets: male, female, young and old patients. Considering an age range between 16 and 98 years old, with a median of 55, a threshold for binarization of young vs. old was set at 50 years. In each subset, a classification task of focal lesional epilepsy vs. non-epileptic was performed.

To keep track of increases or decreases in performance for each classification task, this final ML step of the pipeline was settled to be the same in each iteration. For feature selection, ANOVA scores were used in a univariate selection of K best features; for classification, an SVM model was used; for tuning of hyperparameters, we resorted to a fixed grid search with a wide set of parameters, with Scikit-Learn toolbox’s grid search with cross-validation [43].

For the design of a final model, several different models, together with different methods for feature selection, were trained and assessed, using, once again, Scikit-Learn’s grid search with cross-validation for hyperparameter tuning [43].

## 3. Results

The EEG signal preprocessing, computation of band power, functional connectivity, graph measures and asymmetry ratios resulted in a total of 784 features. The different classification tasks will be presented in this section.

### 3.1. Epileptic vs. Non-Epileptic

In this first approach, ML performance was evaluated with a baseline SVM and ANOVA model.

As one can observe in Figure A3 in the Appendix A, this first classifier achieved an AUC of 0.580 ± 0.036. When comparing the ROC curves, we can conclude that the proposed classifier does not perform better than the surrogate classifiers (mean AUC = 0.584).

### 3.2. Focal Lesional Epileptic vs. Non-Epileptic

Out of all patients with focal lesional epilepsy, CT allowed the identification of 18 cases of trauma, 65 vascular lesions and 50 tumor; with MRI exams, 12 cases of trauma, 62 vascular lesions and 35 tumor were detected in the cohort.

Since focal lesional epileptic patients are a major group in the dataset, when compared to the other types of epilepsy, the focus of this work became to classify focal lesional epilepsy vs. non-epilepsy. The dataset was kept balanced.

#### 3.2.1. Validation Results

First, to confirm whether the presented hypothesis would indeed present better results, the SVM model with ANOVA for feature selection was trained. The results, shown in Figure 3, corroborated our hypothesis, presenting an increase in the AUC of approximately 0.16 when compared to the initial task of classification between epileptic and non-epileptic (Figure A3, in the Appendix A).

It was also speculated that the separation of the data into gender and age subsets could result in an increase in performance. With the SVM and ANOVA model, for the young subset, the mean AUC score was 0.691 ± 0.0894; for the old subset, 0.652 ± 0.0747; for the male subset, 0.674 ± 0.0794; for the female subset, 0.720 ± 0.0550. These mean AUC values, as well as the boxplots in Figure A4, given in Appendix A, show that the model’s performance is globally poorer in the age and gender-focused subgroups.

The ML models were then evaluated on the whole dataset, with classifiers that included SVM, MLP, RFC and LogReg; for dimensionality reduction of feature space, ANOVA scores, PCA and correlation-based feature elimination techniques were implemented. A synthesis of the results in models for the mean and its standard deviation over 5-fold cross-validation can be found in Table 2.

The number of features was optimized for each ML model using the feature reduction techniques discussed in Section 2.8. In Figure A5, given in the appendix, one can find both the training and validation mean scores, through 5-fold cross-validation, for each number of PCA components, in the case of SVM and PCA proposed model. It is visible that two peaks arise: at 3 and 17 components. The peak at 3 PCA components shows a lower difference from training to validation. However, the 17 components were preferred: although for 17 components, the decrease in performance from training to validation is higher, it was considered to give rise to a more stable model with similar validation performance.

From Table 2, the following conclusions were drawn:The SVM using PCA for dimensionality reduction presents a mean AUC score of 0.730 that, while slightly lower than the same model with ANOVA for feature selection, does present better robustness across the different folds, as shown by the lower standard deviation. The accuracy, sensitivity and specificity are also more balanced and globally higher. As for overfitting, it does not seem to be widely overfit, having a difference in AUC score between the training and validation sets of 0.0255.The RFC model shows a mean AUC of 0.745 with a low standard deviation across the five different folds. The values of accuracy, sensitivity and specificity are quite robust too, ranging from 0.699 to 0.730. The major setback of this model is it is overfit to the training set, which presents a difference of 0.254 in the AUC score to the validation set.The LogReg using ANOVA and removal of the most correlated features for feature selection presented the best AUC, accuracy and specificity validation scores. It has a mean AUC score of 0.752, with a standard deviation of 0.038. The mean specificity of 0.795 is quite high, even though its standard deviation is considerable. However, the sensitivity is quite poor and the standard deviation shows great dispersion. Overfit-wise, the values of the AUC score between training and validation present a difference of 0.0173, which indicates a low overfit of the training set.

#### 3.2.2. Test Results

Taking into account both the AUC validation score and the overfitting for each model (SVM, RFC and LogReg), we take the best combination with the feature selection method. Therefore, the three previously highlighted models were submitted to the test set. The test results are summarized in Table 3. Figure 4(i-a–iii-a) shows, in detail, the ROC curves of the models and surrogates, as well as the confusion matrices of the models on the optimal cut-off points. It is relevant to note that, in the case of Figure 4(i-b), in which PCA was used for dimension reduction, only the most important features of each of the 17 principal components are reported, even though each component is a mixture of features. Though the RFC model presents the best AUC score, the SVM and PCA model presents the best accuracy and sensitivity. The LogReg model, with the elimination of highly correlated features and ANOVA for feature selection, had quite a big decrease on the test set, differing approximately 13% from the validation set in the AUC measure. Amongst all models in the test, even though this model presents the best specificity value, globally, it presents the poorest performance.

In general, a decrease in performance is verified from training to test sets, which may be related to the diversity of pathologies in the cohort of patients. The fact that all different models reach a similar performance rather confirms the robustness of the followed approach and points out that the results are not biased by a single choice of model or data reduction method.

#### 3.2.3. Selected Features

Considering the tested models, Figure 4b shows the most important feature types in each model.

From Figure 4(i-b), which reports the features from the best performing model, it is important to mention that the alpha frequency band, as well as the band power features, has not been selected as the most relevant features in any of the 17 PCA components of this model. Furthermore, the use of the global band has greatly surpassed the use of all other frequency bands; graph measures also represent a majority. The use of MI, delta and beta frequency bands, and asymmetry ratios, on each of the groups, respectively, have been the lowest.

From Figure 4(ii-b,iii-b), it is relevant to mention the absence of asymmetry ratios. The RFC model also does not use the theta band. In the LogReg and Corr-ANOVA model, no alpha, theta or beta bands were used, nor were PDC values or band powers. Since the latter model implements the elimination of highly correlated features, the absence of many feature groups may be related to the elimination of correlated features, which are represented in broader features, which use a global band.

In the RFC and the LogReg models, the delta band seems to be among the most relevant frequency bands. Slowing in the EEG is seen in many neurological disorders and seems to be a rather nonspecific sign of a “diseased” brain associated with destructive lesions of cerebral white matter [59,60]. This comes as a result of the fact that our cohort includes patients with different lesions, and there lacks a control group for the lesions as rooting causes for focal lesional epilepsy.

Finally, it is important to highlight that all tested models consistently used IMCOH, PLV and MI as functional connectivity measures, as well as global band-related measures. Graph measures are always the most important feature type, which was expected since they summarize connectivity matrices.

### 3.3. Performance Comparison against Medical First Evaluation of EEG

When comparing the accuracy of the designed models with the accuracy of 0.646 from the doctor’s first EEG analysis (Figure 5), it is possible to conclude that both are similar, with the SVM and PCA model performing a bit better (2.5%) and RFC performing equal to the doctor’s first analysis. This could mean that the EEG does not have enough information for a better initial prediction. On the other hand, while the sensitivity of the SVM and PCA model’s performance is higher than in the doctor’s analysis (by 8.5%), the model we propose in this work is outperformed in terms of specificity (5.8% lower).

The abnormal result flagged by the doctor could be due to pathologies other than epilepsy, such as a tumor brain lesion. As this first diagnosis might lead to incorrect prescriptions of medication, the proposed ML model could be useful as a double-check mechanism to minimize misdiagnosed patients.

Indeed, better results were obtained by combining both inputs–the medical assessment from the first EEG and our best ML model (SVM and PCA). For that, AND and OR logical operators were tested. The performance is summarized in Figure 5. Though the AND operator greatly outperforms our model in terms of specificity, with an increase of 17.2%, both the accuracy and sensitivity significantly decrease (3.7% and 19.2%, respectively). On the other hand, the OR operator presents more unstable performances, with a sensitivity of 0.851 and a specificity of 0.457. Compared to the SVM and PCA model, both the accuracy and sensitivity increase (by 1.2% and 10.6%), while the specificity greatly decreases (11.4%).

## 4. Discussion

In this study, we have built a classifier to diagnose epilepsy from EEG recordings after a suspected first seizure. The dataset consisted of EEG from a cohort of epilepsy patients vs. other pathologies who arrived at an epilepsy unit after a first seizure event. In a focal lesional epilepsy vs. non-epilepsy classification task, the built classifier allies Functional Brain Connectivity-related features to ML, achieving an AUC validation score of 0.730 ± 0.030 and an AUC test score of 0.649. Because of the scarcity of studies on this specific classification task, many different ML models and feature selection schemes were applied and then narrowed down to a selection for further testing.

It is important to highlight that epileptic patients are compared to patients that are not healthy controls. Some of the control patients have brain lesions or other types of pathologies that may affect the recorded EEG, therefore, limiting an epilepsy diagnosis merely based on a first EEG recording. Even though this fact hinders the classification task, this is what the question is in real clinical practice. Compared to other studies in which epileptic vs. healthy patients are predicted, our model presents the possibility of differentiating patients with and without epilepsy.

### 4.1. Focal Lesional Epilepsy-Based Classification

The best performance for our classification task is achieved by the model that focuses on focal lesional epileptic patients. Since this represents the most common type of epilepsy present in the dataset (Figure 1), and as each epilepsy type will have different EEG characteristics [13], an ML model focused on a specific type of epilepsy is expected to perform better than one that aims to classify a more general problem.

Furthermore, as previously mentioned, significant differences in connectivity measures linking distinct brain regions between focal epileptic subjects and control subjects on resting-state EEG without interictal epileptiform discharges have been previously shown [18], which likely also contributes to a good performance on the focal lesional classification model. This is also supported by the presence of functional connectivity features, specifically IMCOH, MI and PLV, as well as their graph measures across all tested models.

The use of the global band across all tested models (Figure 4(i-b–iii-b)) is also interesting, as it might mean that the wider frequency band conveys a better synthesis of the information contained in the full frequency spectrum.

While differences in band power between healthy and epileptic patients on resting state EEG recordings have been reported [61], the SVM and PCA model, which presented the best test results, did not use band power features for the classification task, as shown in Figure 4(i-b). Nevertheless, both the RFC and LogReg and Corr-ANOVA models, also subjected to testing, did use the band power features for the classification task, Figure 4(ii-b,iii-b). Depending on the ML models, the band power features do hold some predictive power and can be a useful addition to the pipeline. As verified in the RFC and LogReg models, delta-related features play a considerable role. Cerebral lesion often results in delta band slowing as an unspecific sign of the lesion, which highlights the fact that a control group to assess the effect of each type of lesion could be relevant in a future study.

The importance of asymmetry ratios is not fully clear since they are only used in the SVM and PCA model.

### 4.2. Performance Comparison against Medical First Evaluation of EEG

Globally, these results match the expectations. Considering the truth tables, an AND operator should increase the non-epileptic predictions (negative outputs), while an OR operator should increase the epileptic predictions (positive outputs). Taking into account that the increase in epileptic predictions results in increased sensitivity, the OR operator could be especially interesting in the context of epilepsy diagnosis at the suspected first seizure. This way, both the ML model’s output and the doctor’s opinion would be considered towars a more informed decision. Follow-up tests such as MRI or long-term EEG exams can be performed to confirm the diagnosis so that not too many patients are missed.

### 4.3. Feature Elimination Based on Correlation

Even though this technique produced the best validation scores, it lacked robustness, with the worst performance on the test set (AUC score of 0.624). However, since the elimination of features is performed randomly (i.e., after assessing that two features are correlated, one of them is randomly chosen to be eliminated), in the future, it would be of interest to implement an algorithm that would choose the feature of most interest for the classification task using, for example, the *p*-value.

### 4.4. Future Perspective

Future work could include the use of new functional connectivity measures or deep learning techniques, which have been showing good performances in state-of-art epilepsy studies [62]. Besides that, a more thorough analysis of the dataset’s metadata could be of added value, for instance, by segregating subjects with different brain lesions, epilepsy drugs or family history. It could also be interesting to investigate our model’s misclassifications, find a pattern and design specific approaches to tackle each mistake.

Given these results and the heterogeneity of epileptic disorders, future work will have to reveal whether an ensemble of classifiers trained specifically for different epilepsy types would perform better than one generalized model.

Regarding the EEG preprocessing, the effectiveness of using 40 Hz as the upper limit of the pass-band filter should be verified, as there may be some attenuation on the upper part of the beta frequency band (13–30 Hz). Moreover, there could be a more thorough investigation of the effect of including or excluding fragments with hyperventilation and photic stimulation.

As this study did not include a control group of lesional patients without seizures, future studies are needed to prove that the delta frequency band can be a reliable marker to differentiate lesional patients with epilepsy from those without epilepsy.

Finally, the design of an additional classifier looking for brain lesions regardless of the diagnosis could be interesting to assess the importance of brain lesions in a classification based on a suspected first seizure EEG. Regarding the medical evaluation of EEG, it would also be interesting to separate the patients with different EEG labels into two groups (normal and abnormal) and, once again, repeat the performed classification task in each of these groups. This could, once again, help the doctor confirm their preliminary diagnosis and proceed in the best possible way.

## 5. Conclusions

The classification of focal lesional epileptic patients showed better performance, as the classification is based upon a more specific EEG pattern. The choice for the focal lesional group was based on the fact that not only is it the most represented group in the dataset but also that previous studies have shown abnormalities in focal epileptic patients’ EEG. The model that produced the best results was the SVM using PCA for dimensionality reduction, with an AUC of 0.649, an accuracy of 0.671, a sensitivity of 0.745 and a specificity of 0.571. This model performed equally when compared with the accuracy of the doctor’s classification of the EEG as normal or abnormal at the first analysis. This result indicates that there is considerable predictive power for the diagnosis of epilepsy—a structural and functional network disorder—on functional connectivity, particularly robust on IMCOH, PLV and MI, and its graph measures, which summarize connectivity matrices. The global band features also had a good performance, being significantly represented in all tested models.

With some more refinement, the proposed classifier may be interesting as an aiding mechanism in a first EEG evaluation, aiming to minimize the rate of epilepsy misdiagnosis.

## Figures and Tables

**Figure 1 bioengineering-09-00690-f001:**
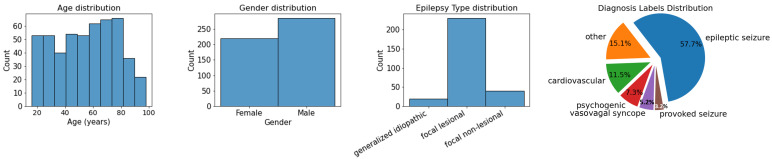
Metadata distribution regarding age, gender and epilepsy type. In the latter category, only patients with epilepsy diagnoses are included. On the right, the pie chart represents metadata distribution regarding clinical diagnosis, confirmed more than 1 year after the suspected first seizure.

**Figure 2 bioengineering-09-00690-f002:**
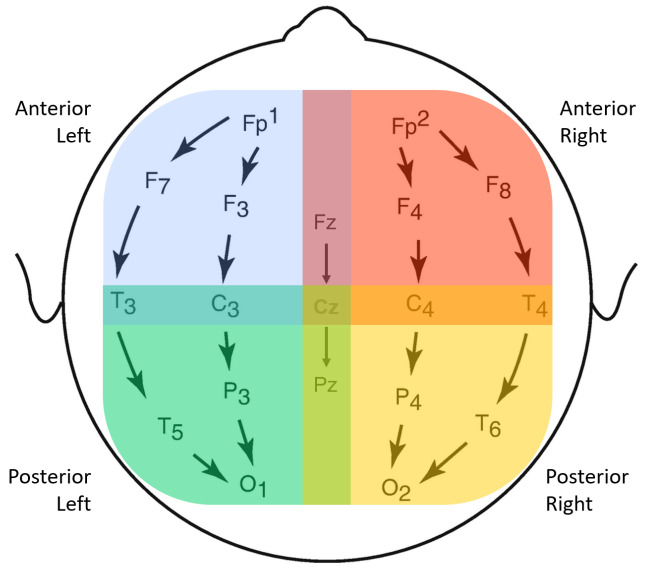
Longitudinal Bipolar Montage set to the 19 original EEG channels and the 4 channel groups, separated according to spatial locations. A total of 18 bipolar sensor pairs are considered. In the feature extraction stage, features are first extracted from each single channel and then averaged over each group.

**Figure 3 bioengineering-09-00690-f003:**
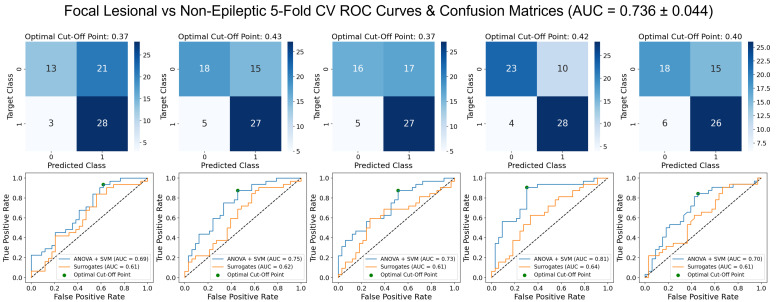
ROC curves and optimal confusion matrices for 5-fold cross-validation in focal lesional epileptic (labeled as 1) vs. non-epileptic (labeled as 0) classification, with SVM and ANOVA for feature selection. Each plot (of the five) represents one fold.

**Figure 4 bioengineering-09-00690-f004:**
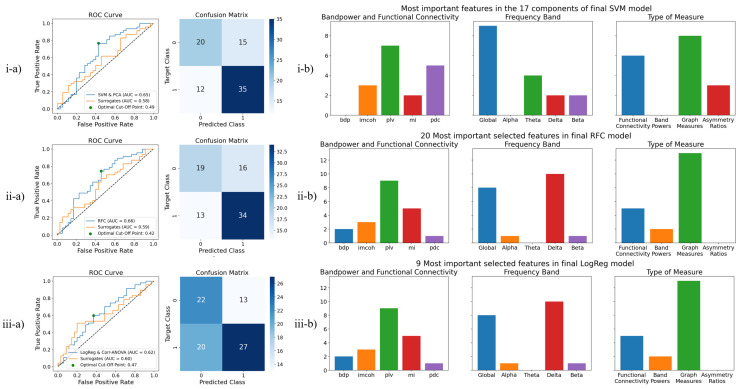
Final models for the classification of focal lesional epileptic (labeled as 1) vs. non-epileptic patients (labeled as 0), with a report of test results and final used features. From top to bottom, the included models are: (**i**) SVM and PCA for dimensionality reduction; (**ii**) RFC; (**iii**) Logistic Regression with ANOVA scores for feature selection and correlation-based elimination of features. From left to right, one can find the (**a**) ROC Curve and Confusion Matrix for the test set, in optimal cut-off point; (**b**) the analysis of final features, in which there is a report of the features’ origin, the frequency band considered in the feature or the type of measure.

**Figure 5 bioengineering-09-00690-f005:**
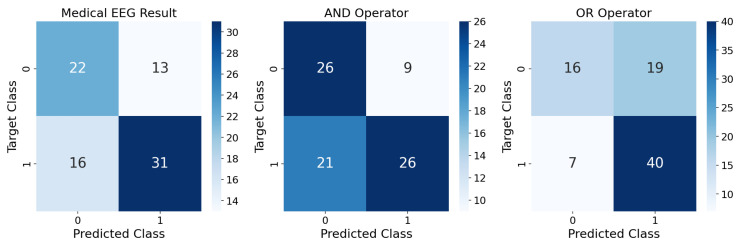
In the left confusion matrix, the performance of medical evaluation (metadata’s “EEG Result”) based on a first evaluation of the EEG signal is compared with the actual diagnosis two years later. The test set is used for focal lesional epilepsy (labeled as 1) vs. non-epileptic classification (labeled as 0). The accuracy of this first medical evaluation is 0.646, the sensitivity is 0.660 and the specificity is 0.629. In the mid confusion matrix, the result of an AND operator between the medical diagnosis from the first EEG and our best algorithm (SVM and PCA) is assessed. The accuracy is 0.634, the sensitivity is 0.553 and the specificity is 0.743. In the right confusion matrix, the result of an OR operator between the EEG result and the ML prediction is shown. The accuracy achieved is 0.683, the sensitivity is 0.851 and the specificity is 0.457.

**Table 1 bioengineering-09-00690-t001:** Synthesis of extracted features.

Type of Feature	Feature	Comments
Band Power	Delta (1–4 Hz) Theta (4–8 Hz) Alpha (8–13 Hz) Beta (13–30 Hz) Total Absolute Power (1–30 Hz)	Epochs combined with median and std. Scalp groupings: AL, AR, PL, PR.
Functional Connectivity	Imaginary Part of Coherency (IMCOH) Phase-Locking Value (PLV) Mutual Information (MI) Partial Directed Coherence (PDC)	Epochs combined with mean, except for MI, which combines epochs with median, and PDC that combines epochs in MVAR. Five different frequency bands. Scalp groupings: AL, AR, PL, PR, spatially combined with mean and standard deviation.
Graph Measure	Efficiency Betweenness Centrality Clustering Coefficient Node Strength In and Out Degrees (directed, PDC only)	Epochs combined with mean, except for MI, which combines epochs with median, and PDC that combines epochs in MVAR. Five different frequency bands. Scalp groupings: AL, AR, PL, PR, spatially combined with mean and standard deviation.
Asymmetry Ratios	Anterior: left vs. right (AL vs. AR) Posterior: left vs. right (PL vs. PR)	Based on graph measures, which summarize connectivity matrices. Ratios only relative to mean values from Betweeness Centrality, Clustering Coefficient and Node Strengths; maximum value from In and Out degrees.

**Table 2 bioengineering-09-00690-t002:** Synthesis of the 5-fold cross-validation classifiers for focal lesional epilepsy vs. non-epilepsy classification, with AUC, accuracy, sensitivity and specificity assessment metrics. In a validation vs. training performance check, we report the difference between AUC scores in training and in validation.

#	Model	AUC	Accuracy	Sensitivity	Specificity	Val vs. Train
1	SVM (linear) and ANOVA	0.736 ± 0.044	0.689 ± 0.052	0.856 ± 0.031	0.531 ± 0.102	0.0426
2	SVM (gaussian) and PCA	0.730 ± 0.030	0.705 ± 0.020	0.692 ± 0.084	0.717 ± 0.061	0.0255
3	SVM (linear) and Corr-ANOVA	0.735 ± 0.037	0.689 ± 0.034	0.672 ± 0.162	0.704 ± 0.118	0.0345
4	MLP and ANOVA	0.732 ± 0.047	0.717 ± 0.051	0.749 ± 0.116	0.688 ± 0.117	0.0082
5	MLP and PCA	0.734 ± 0.026	0.698 ± 0.018	0.674 ± 0.141	0.724 ± 0.141	0.0576
6	RFC	0.745 ± 0.022	0.714 ± 0.012	0.730 ± 0.069	0.699 ± 0.048	0.254
7	RFC and ANOVA	0.718 ± 0.047	0.686 ± 0.050	0.767 ± 0.080	0.609 ± 0.161	0.282
8	RFC and PCA	0.706 ± 0.022	0.698 ± 0.023	0.655 ± 0.117	0.742 ± 0.140	0.272
9	LogReg and ANOVA	0.734 ± 0.048	0.698 ± 0.057	0.775 ± 0.209	0.629 ± 0.205	0.0344
10	LogReg and PCA	0.729 ± 0.033	0.698 ± 0.023	0.704 ± 0.107	0.693 ± 0.131	0.0028
11	LogReg and Corr-ANOVA	0.752 ± 0.038	0.717 ± 0.037	0.634 ± 0.189	0.795 ± 0.153	0.0173

**Table 3 bioengineering-09-00690-t003:** Assessment metric results for the test set, in final models for focal lesional epileptic vs. non-epileptic patients.

Final Model	AUC	Accuracy	Sensitivity	Specificity
SVM and PCA	0.649	0.671	0.745	0.571
RFC	0.655	0.646	0.723	0.543
LogReg Corr-ANOVA	0.624	0.598	0.574	0.629

## Data Availability

Not applicable.

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
