# Peer review of "Diagnosis of Epilepsy with Functional Connectivity in EEG after a Suspected First Seizure"

_bioengineering, 2022, doi:10.3390/bioengineering9110690_

Round 1

Reviewer 1 Report

Matos present the application of machine learning techniques to EEG data to develop a "classifier to diagnose epilepsy from EEG recordings after a first seizure". The idea is interesting and worthy of study, but I feel there are important clarifications required.

CLINICAL ASPECTS

Whilst the target journal is technical, there are significant errors or omission in the clinical aspects of the paper. The most fundamental is that is unclear how the predicted outcome that the classifier is using is determined which is critical to the understanding of the results. It appears that some degree of followup was performed, but how is somebody classified as having epilepsy or not?

The cohort is described is those presenting with "first seizure" but the pie chart in Figure 1 shows that the diagnosis for many was of something other than a seizure, so presumably it was those presenting with a SUSPECTED first seizure. The introduction refers to a diagnosis of epilepsy with two unprovoked seizures and refers to criteria from 1989 (reference 8). These have been long superseded with the current diagnostic criteria available at https://www.ilae.org/guidelines/definition-and-classification. Of particular note is that epilepsy can be diagnosed after a SINGLE unprovoked seizure if risk of recurrent is considered high (such as there being a lesion).

Other errors include "a correct disease diagnosis can lead to a better quality of life" (abstract) - this does not follow, it is the successful treatment that can do this; "Epilepsy is a neurological disorder, characterized by recurrent seizures, which may be 20 brief episodes that involve part or the entirety of the body, caused by an interruption of loss or alteration of consciousness" (introduction) - this is incorrect, the alteration of consciousness is part of the disorder not the cause of the episodes; vasovagal syncope (not vagal syncopes); lumbar puncture is NOT a typical diagnostic tool for epilepsy - EEG and MRI are the key tools.

INTRODUCTION

There is other important literature in this type of cohort (e.g. https://pubmed.ncbi.nlm.nih.gov/20520774, https://pubmed.ncbi.nlm.nih.gov/23565166) which does not seem to have been taking into account in the authors literature review so the novelty needs to be better expressed with reference to these and any other studies. References need to be double checked (e.g. references 16 and 17 are the same).

METHODS

Patients with sleep were excluded, but many patients become drowsy or sleep during an EEG. Do you only exclude those with confirmed stage N2 sleep on EEG? Also "sleep-sate EEG" (typo). The dataset is reduced from 629 to 504 patients - state the number of exclusions for each of the reasons.

For the EEG preprocessing, the upper pass filter of 40Hz is too low if you wish to analyse power in the beta frequency band (13-30Hz) since significant attenuation will apply at the upper part of this range.

I do not follow why a longitudinal bipolar montage is used to assess connectivity since the comparison of adjacent electrodes may lead to cancellations artefacts potentially precluding such analysis. The two studies I referenced above use a referential montage for example. Is there any literature to support the validity of this approach for connectivity analyses?

RESULTS

The results present a large number of ML models and data reduction approaches, without providing any clear justification of why these specific approaches were chosen. This risks making it sound like a we threw everything but the kitchen sink approach.

In the results, the authors change to classifying the large group (focal lesional) vs non-epileptic without clearly explaining this approach. No details are given of the lesions present and this turns the study in one trying to classify something which is defined by another modality (i.e. imaging) by a completely different modality. The clinical relevance needs to be clearer.

Author Response

Thank you for your helpful remarks, we have included them as well as we could! Your comments helped us to better explain some of the technical details, as well as the interpretation of the results and clinical context of our work. We hope you are satisfied with the updated manuscript. Please our response to your remarks in the attached document.

Reviewer 2 Report

The manuscript on hand describes the application of functional connectivity metrics on resting state scalp EEG. The authors used a broad range of well selected metrics from different toolboxes in a relevant group of patients mainly suffering from focal epilepsy. After dimensionality reduction of the feature space the authors compare different machine learning classifiers to combine features. The performance of their methods shows moderate results comparable with ratings of clinical evaluators, but it needs to be taken into account that they seem to have evaluated only brief EEG recordings. The discussion includes the most relevant topics. The manuscript is well written. I suggest to reconsider it for publication after major revision.

Title: ” Diagnosis of Focal Lesional Epilepsy with Functional Connectivity in EEG after a First Seizure

Although the authors mainly included patients with focal lesional epilepsy restricting the title to this group of patients is not completely corrected as the authors also included patients from other groups. I suggest to at least remove the term ‘lesional’ from the title.

p.1 Introduction:

“Epilepsy is usually diagnosed after a person has experienced at least two unprovoked seizures, not caused by a known condition like alcohol withdrawal or extremely low blood sugar [3].”

In the context of the manuscript this is not the most important information. Epilepsy can be also diagnosed after a single unprovoked seizure when there is an increased recurrence risk e.g. by having a potentially epileptogenic lesion or epileptic discharges in EEG [Pohlmann-Eden et al., 2006]. As the authors most often seem to have included patients with potentially epileptogenic lesions and a seizure, those patients very likely have the diagnosis of epilepsy. The authors should make this clear throughout the manuscript especially in the introduction and the conclusion so that it remains consistent with the title.

p.3, 2.1. Dataset description

The authors need to describe how they selected the EEGs. Was it a continuous group of patients? During which time window they included EEGs? What was the setting in which the patients had the EEG? What was the duration of EEG recordings? Did the authors analyse data sets with hyperventilation and photic stimulation? Were the EEG electrodes glued as in video EEG monitoring to the scalp or was it routine EEG recordings?

Figure 1: Diagnostic labels: This panel is confusing as it is not clear, if those patients with ‘epileptic seizures’ suffer from epilepsy. I assume that the authors selected this group as the group of patients with epilepsy and the other ones as control group. Also later throughout the manuscript the authors have to state clearly, if those patients had the diagnosis of epilepsy.

p.4, 2.5 Epoch selection

Please state why you selected the number of 50 epochs!

p.5

2.6.1. Band power features

Since there has been a description of alterations on EEG’s band powers during different epileptic states [30,31]. The author might consider mentioning that there are other studies which identified power changes in different frequency bands e.g. [Heers et al., 2018]

p.13 figure 4: Why did the authors skip reporting results of the Multi-layer Perceptron (MLP) here?

In the RFC and the LogReg model the delta-band seems to be among the most relevant frequency bands. In cerebral lesions there is often delta slowing as unspecific sign of a cerebral lesion. For me it looks like the authors need to discuss the problem that they have a huge group of patients with lesions and a single seizure. As they don’t have a control group of lesional patients without seizures they need to mention, that future studies are needed to proof that the delta frequency band is a reliable marker differentiate lesional patients with epilepsy from those without epilepsy.

p.15, 4.2. Performance comparison against medical first evaluation of EEG

It is somewhat untypical to put a new comparison in the discussion. The authors should consider adding this to the results section and discuss it in the discussion section.

Minor comment: there is a typo ‘the the’ on page 15.

References:

Heers M, Helias M, Hedrich T, Dümpelmann M, Schulze-Bonhage A, Ball T (2018): Spectral bandwidth of interictal fast epileptic activity characterizes the seizure onset zone. NeuroImage Clin 17:865–872.

Pohlmann-Eden B, Beghi E, Camfield C, Camfield P (2006): The first seizure and its management in adults and children. BMJ 332:339–342.

Author Response

Thank you for your helpful remarks, we have included them as well as we could! Your review definitely helped us to better explain the clinical context of this work and include some missing points in the discussion. We hope you are satisfied with the updated manuscript.

Round 2

Reviewer 2 Report

The authors adressed all points that were raised during the review. It is a relevant contribution to the analysis functional connectivity of EEG data from a first seizure clinic for the diagnosis of epilepsy.

I can fully support its publication in the journal bioengineering.

Minor comment:

p2: Generalized seizures: originated at some point in the brain, from where a widespread electrical discharge rapidly involves the entire brain

I bleive it should be widespread 'epileptic' discharge

Author Response

Dear reviewer,

Thank you for you final comment and for endorsing our paper. We have made the final adjustment and changed "electrical discharge" to "epileptic discharge".

Kind regards